# Imbalanced Tabular Data Synthesis via LLM-Seeds and Interpolation

## Abstract

Imbalanced tabular data poses a persistent challenge in machine learning, as classifiers often underperform on the minority-class data when trained on skewed data. Synthetic minority data generation is a standard approach, with traditional interpolation-based methods offering efficiency but limited representational capacity and being inapplicable under severe imbalance due to their reliance on existing data points. Recent approaches that employ large language models (LLMs) address this limitation by generating synthetic samples informed by contextual knowledge, but they are computationally expensive and often impractical at scale. To bridge this gap, we propose a hybrid approach that integrates the strengths of both strategies: LLMs generate a small set of contextually meaningful seed samples to expand the observed minority support, while interpolators efficiently generate additional samples within this augmented support. As a foundational work on this hybrid approach, we use standard LLMs and interpolators for experiments to observe better the benefits of the hybrid design and provide baseline results for further research. Extensive experiments across 60 benchmark tabular datasets show that the hybrid approach provides considerable efficiency gain over the LLM-only method without performance degradation, demonstrating the potential of the hybrid approach as a complementary strategy for synthetic minority data generation in imbalanced tabular learning.

## 1 Introduction

Learning from imbalanced tabular data is a long-standing challenge in many real-world machine learning applications, class distributions are highly skewed, leading standard classifiers to favor the majority class and underperform on the minority class (Chen et al. (2024)). Synthetic data generation, which creates additional instances of the minority class, is widely used to address this problem (Panagiotou et al. (2024)). Traditional interpolation-based methods (or interpolators), such as Synthetic Minority Oversampling Technique (SMOTE) (Chawla et al. (2002)), generate synthetic samples by interpolating between existing minority instances. While effective and computationally efficient, they are fundamentally limited: they can only operate within the existing data instances; their representation is limited and they may overfit or fail altogether when minority samples are scarce or absent (Li et al. (2025); Khorshidi & Aickelin (2025)).

In contrast, large language models (LLMs) have opened new possibilities for synthetic data generation. Unlike interpolators, LLMs can leverage their broad contextual knowledge to generate synthetic data that is not constrained by the empirical distribution of existing data. Recent studies show that LLMs can produce plausible (i.e., well-represented) tabular data (Borisov et al. (2022); Kim et al. (2024); Long et al. (2025)). However, such LLM-only methods for tabular data synthesis are typically computationally expensive, requiring multiple prompt–generation cycles and significant resources (Nguyen et al. (2025); Sui et al. (2024)). This limits their practical availability, particularly for large-scale or resource-constrained applications. Although some studies addressing the efficient use of LLMs with tabular data have been conducted (Wu & Hou (2025)), efficient tabular data generation while maintaining LLM-comparable performance remains underexplored.

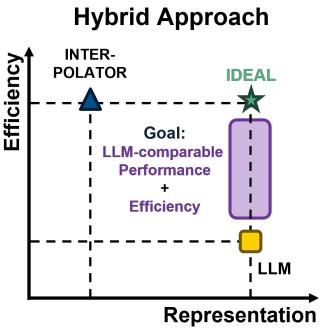

Figure 1: A hybrid approach enables high-quality representation with greater efficiency.

To address this gap, we propose a hybrid approach that combines the complementary strengths of LLMs and interpolators (e.g., SMOTE) as shown in Figure 1. Our goal is the purple area: LLM-comparable performance (i.e., high-quality representation) with greater efficiency than the LLM-only method. Specifically, *LLMs expand the observed minority support by generating a small set of contextually meaningful seed samples, and additional synthetic data samples are generated by interpolation in the augmented area.* This hybrid approach combines the representational power of LLMs with the efficiency of interpolation.

To the best of our knowledge, we first try this hybrid design leveraging complementary strengths of LLMs and interpolation. As a foundational work, this paper provides theoretical analyses of how and why this approach works, and provides extensive experimental results to demonstrate its benefit. Specifically, hybrid methods remain functional under few-shot and zero-shot conditions where interpolators are inapplicable, have more performance gains over other methods as imbalance becomes more severe, and show significant efficiency gains over the LLM-only methods.

The contributions: 1) We introduce a novel hybrid approach, which reciprocally uses LLMs and interpolators, followed by theoretical and empirical analysis. 2) We provide baseline results with standard models for future research, not only demonstrating the effectiveness but also providing insights. 3) Extensive experimental settings with many real-world datasets improve the robustness and generality of the results.

## 2 Related Work

Research on synthetic tabular data generation has spanned a broad spectrum, from traditional interpolation-based methods to advanced approaches that leverage LLMs. We provide a brief review of the two categories, afterward, introduce the capabilities of LLMs as data generators.

### 2.1 Interpolation-based Data Generation

Interpolation-based data generation methods (interpolators) are widely used to address class imbalance, and SMOTE, as one of prominent methods, generates synthetic minority samples by interpolating existing data samples. A number of SMOTE variants have been proposed (Kovács (2019)), including Borderline-SMOTE (Han et al. (2005)) or ADASYN (He et al. (2008)), which attempt to refine the generation process by focusing on decision boundaries or adaptively weighting instances. Despite their popularity and efficiency, all interpolators share a common limitation: they entirely rely on the distribution of available minority samples (Li et al. (2025)). When the number of minority points is scarce, they may generate redundant or uninformative data, and in extreme cases, they may not operate at all (Khorshidi & Aickelin (2025)). For instance, they are inapplicable in a zero-shot setting due to the lack of existing reference data points.

### 2.2 LLM-based Data Generation

Recent studies have explored using LLMs to generate tabular datasets by framing the generation process as a text-to-table task. GreaT (Borisov et al. (2022)) is considered one of early works in the LLM-based tabular data generation area, using finetuning for a pretrained LLM with permutation. More recent works, including Epic (Kim et al. (2024)), HARMONIC (Wang et al. (2024)), and LLMOverTab (Isomura et al. (2025)), have been introduced, meaning the efforts leveraging LLMs as tabular data generators has been active. In addition, another work shows that LLMs are capable to generate custom-designed patient data (Törnqvist et al. (2025)), demonstrating the practical use case. These approaches leverage LLMs' contextual understanding and extensive knowledge base to generate diverse, semantically meaningful synthetic samples. However, these methods incur a high computational cost (Nguyen et al. (2025); Sui et al. (2024)). Generating synthetic data with LLMs often requires multiple prompt–response cycles and significant resource consumption, making them impractical in large-scale or resource-constrained environments (Chan et al. (2024)).

### 2.3 LLM Capability

We highlight LLM capabilities that are important in some unique cases. Transitioning data with table-to-text flow may lose complicated relationships between features. One work addressed this (Long et al. (2025)), although LLMs are indirectly used in this method for compressing tabular data, this shows that LLMs are capable for capturing the inter-column logical relationship and preserving it. Other recent works show that LLMs are applicable in a zero-shot setting, where no reference data samples exist. Ye et al. introduced a zero-shot learning method via dataset generation, demonstrating LLMs' generation capability without reference examples (Ye et al. (2022)). In another work, LLMs are used to generate interaction data for unknown users and objects in sequence for zero-shot recommendation (Peng et al. (2026)). These works show LLMs' capabilities as tabular generators, highlighting that they can even perform in harsh conditions, such as complicated inter-column relationships and low-reference data.

## 3 Hybrid approach: LLM-Seeds followed by Interpolation

Prior work has largely treated interpolation and LLM-based data generation as separate directions (Shi et al. (2025)). Hybrid approaches that explicitly combine the efficiency of interpolation with the representational power of LLMs remain unexplored. The proposed hybrid approach fills this gap leveraging LLMs to generate a small yet well-represented set of seed samples and then employs interpolators to efficiently fill the augmented area. This design aims to achieve LLM-comparable performance while maintaining efficiency.

### 3.1 Operation of the Hybrid Approach

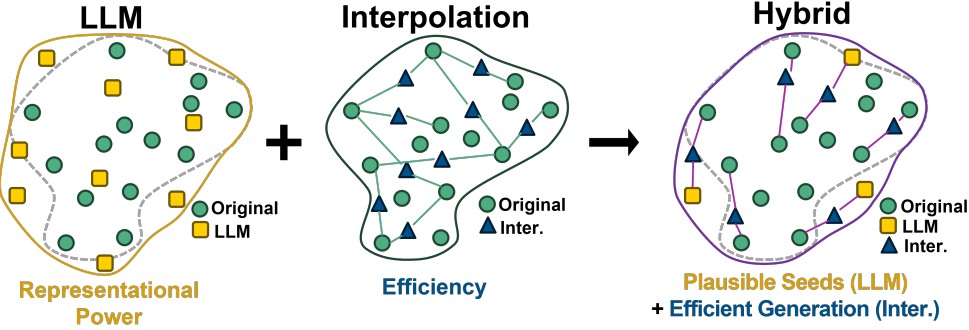

Figure 2: Conceptual implementation of the hybrid approach, combining LLM's representational power (left) and efficient interpolation (center). In the hybrid approach (right), an LLM expands the observed minority support by generating contextually meaningful seed samples (yellow squares). An interpolator efficiently generates more samples in the augmented support (navy triangles).

The conceptual implementation of the hybrid approach is described in Figure 2. The left (LLM) shows that the LLM expands the observed minority support (gray dashed → yellow solid line), independent of the current data points. The center (Interpolation) shows that an interpolator generates synthetic data points relying on the current data points. The right (Hybrid) shows the hybrid approach. An LLM generates a small set of contextually meaningful minority seeds (yellow squares), expanding the observed support of the minority class (purple solid line). Then, an interpolator generates additional synthetic samples (blue triangles), filling in the augmented support with computationally inexpensive interpolation. LLMs enable the exploration of data beyond the observed support, which is particularly valuable in extremely imbalanced scenarios. At the same time, interpolators ensure scalability by handling the bulk of data generation at negligible computational cost. By invoking the LLM only for seed generation, the hybrid approach avoids the repeated overhead of LLM-only methods while preserving their representational benefits. The hybrid approach achieves both effectiveness and efficiency, making it suitable for practical applications.

**Note That:** Although any capable LLMs and interpolators can be used for the hybrid approach, in this paper, we employ GPT-4o-mini (GPT) (OpenAI (2024)) and Devstral 2 (DEV) (MistralAI (2025)) as the LLM component, and SMOTE for the interpolator component. GPT and DEV are not specifically designed for tabular data generation, and SMOTE is widely recognized as the standard interpolator (Fernández et al. (2018)) without additional heuristics. By adopting these standard LLMs and SMOTE, we can more clearly observe the benefits of the hybrid design rather than those of designated LLMs or advanced SMOTE-variants. As a foundational work of the hybrid approach, this paper can serve as a baseline for further research. Our goal is not to benchmark different combinations but to demonstrate the effectiveness of the hybrid approach.

### 3.1.1 Formulation

Since each LLM-based method operates differently, we provide the general formulation for LLMs as shown in Eq.(1). $\mathcal{P}$ is the prompt template, $\mathcal{D}$ represents the input data, and $\theta$ indicates LLM parameters. Details about our pipeline with prompts are provided in the Appendix A.1.

$$\mathbf{x}_{\mathrm{syn}} = f_{\mathrm{LLM}}(\mathcal{P}, \mathcal{D}, \theta) \tag{1}$$

Although each interpolator has different data generation process, the interpolation can be explained by the standard SMOTE as Eq.(2). SMOTE generates synthetic minority class samples by interpolating between existing minority class samples, where $\mathbf{x}_i (\in S_{\mathrm{min}})$ is a minority sample and $S_{\mathrm{min}}$ denotes the minority class sample set. For each minority instance $\mathbf{x}_i$, SMOTE selects $\mathbf{x}_{nn(i,j)}$, its $j$-th nearest neighbor of $\mathbf{x}_i$ in the minority class, and generates new samples along the line segments connecting the instance to its neighbors.

$$\mathbf{x}_{\mathrm{syn}} = f_{\mathrm{INTER}}(\mathbf{x}_i) \approx f_{\mathrm{SMOTE}}(\mathbf{x}_i) = \mathbf{x}_i + \lambda \cdot (\mathbf{x}_{nn(i,j)} - \mathbf{x}_i), \quad \lambda \sim U(0,1) \tag{2}$$

Let $N_{\mathrm{maj}}$ be the majority class size, $N_{\mathrm{min}}$ be the original minority class size, and $r_{\mathrm{target}}$ be the target minority ratio. The total number of synthetic samples needed is:

$$N_{\mathrm{syn}} = N_{\mathrm{maj}} \cdot r_{\mathrm{target}} - N_{\mathrm{min}} \tag{3}$$

For each LLM- and interpolator-only method, the augmented minority set is:

$$\begin{cases} S_{\mathrm{min}}^{\mathrm{LLM}} = S_{\mathrm{min}} \cup S_{\mathrm{LLM}} = S_{\mathrm{min}} \cup \left\{ f_{\mathrm{LLM}}(\mathcal{P}, \mathcal{D}, \theta) \right\}_{i=1}^{N_{\mathrm{syn}}} \\ S_{\mathrm{min}}^{\mathrm{INTER}} = S_{\mathrm{min}} \cup S_{\mathrm{INTER}} = S_{\mathrm{min}} \cup \left\{ f_{\mathrm{INTER}}(S_{\mathrm{min}}) \right\}_{i=1}^{N_{\mathrm{syn}}} \end{cases} \tag{4}$$

In the hybrid approach, the samples are allocated between LLM and the interpolator as follows, where $\mathrm{r}_{seed}$ is the ratio for LLM to generate seed:

$$\begin{cases} N_{\mathrm{LLM}} = N_{\mathrm{maj}} \cdot r_{\mathrm{seed}} - N_{\mathrm{min}} \\ N_{\mathrm{INTER}} = N_{\mathrm{syn}} - N_{\mathrm{LLM}} \end{cases} \tag{5}$$

For the hybrid approach, the final augmented minority set is:

$$\begin{aligned} S_{\mathrm{min}}^{\mathrm{HYBRID}} &= S_{\mathrm{min}} \cup S_{\mathrm{LLM}} \cup S_{\mathrm{INTER}} \\ &= S_{\mathrm{min}} \cup \left\{ f_{\mathrm{LLM}}(\mathcal{P}, \mathcal{D}, \theta) \right\}_{i=1}^{N_{\mathrm{LLM}}} \cup \left\{ f_{\mathrm{INTER}}(S_{\mathrm{min}} \cup S_{\mathrm{LLM}}) \right\}_{i=1}^{N_{\mathrm{INTER}}} \end{aligned} \tag{6}$$

### 3.1.2 Seed Ratio

Multiple seed ratios ($\mathrm{r}_{seed}$) are available (by user selection), which decides the number of samples generated by the LLM. For detailed explanation, the application of the hybrid method using an example dataset is provided in Appendix A.2. It is important to note that a smaller seed ratio enables greater efficiency (reduced LLM use) and raises the chance of better performance (more target-ratio options generate more augmented datasets). Therefore, a smaller seed ratio may be a reasonable choice; however, a smaller LLM portion may degrade performance by limiting the LLM's representational power. So this questions the potential efficiency-performance trade-off when choosing the seed ratio. To investigate this, we conduct a seed-ratio sensitivity analysis in Section 6.

### 3.2 Quality of LLM-Seeds

Although we reviewed recent related works supporting LLMs' capability as synthetic tabular data generators in Section 2, in this section, we provide quantifiable demonstration with the employed LLMs in this paper. As seed samples for further data generation, LLM-generated samples need two capabilities: 1) they should be able to expand the observed minority support, and 2) they should be informative for classification. To show these capabilities, we measure k-nearest neighbor (kNN) distance and classification uncertainty of generated data samples by the used LLMs and SMOTE across 60 datasets (Table 9 in the Appendix A.7). For the measurement, we employ the first target ratio strategy (S1) with all target ratios as {0.2, 0.4, 0.6, 0.8, 1.0} on the train data (70%) of each dataset.

#### 3.2.1 Expandability

To quantify if LLM-generated samples expand the observed minority support, we measure the mean kNN (k = 5) distance from synthetic samples to the nearest real minority samples across 60 datasets. We compare GPT vs. SMOTE and DEV vs. SMOTE. Figure 3 presents a paired comparison between SMOTE-generated (X-axis) and LLM-generated (Y-axis) samples, where each point corresponds to the mean kNN distance of samples in a dataset (i.e., a total of 60 points). A log-scale is used to accommodate large differences and to ensure readability in the presence of a few extreme values. Points located above the diagonal indicate that LLM-generated samples lie farther from the observed minority samples than those generated by SMOTE. With both LLMs, we observe that the majority of datasets (GPT = 59 and DEV = 57) are located above the diagonal, indicating that LLMs produce samples that extend beyond the interpolation by SMOTE. In some datasets, the kNN distance of LLMs are farther than others, implying potential noisy data points (i.e., data located too far from the original support may be irrelevant data). Nevertheless, the results show that LLM-generated seed samples provide expansion from the observed minority support.

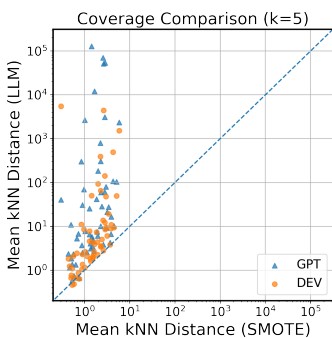

Figure 3: Coverage comparison between SMOTE and two LLMs. Both LLMs generate farther data points than SMOTE.

#### 3.2.2 Informativeness

To evaluate whether LLM-generated samples contribute to learning the decision boundary, we measure classifier uncertainty, defined as the absolute deviation of predicted probabilities from 0.5 as shown in Eq. (7).

$$\textbf{Uncertainty}(\mathbf{x}) = |\mathbf{P}(x) - \mathbf{0.5}| \tag{7}$$

This measures how close the generated sample is located from the decision boundary. Samples near the decision boundary are more informative for classification because they help the classifier learn class separation, reducing ambiguity (Han et al. (2005)). We use three nonlinear classifiers: Support Vector Machine (SVM), kNN, and Decision Tree (DT). Figure 4 presents paired comparisons between SMOTE and LLM-generated samples across 60 datasets with the three classifiers in order. As interpretation, points located under the diagonal indicate that LLM-generated samples have less uncertainty (i.e., more informative).

For SVM, LLM-generated samples show lower mean uncertainty (i.e., they are closer to the decision boundary) than SMOTE-generated samples in 42/60 (GPT) and 46/60 (DEV) datasets, indicating that they provide more informative samples for learning the decision boundary. For kNN, they mostly provide the same pattern, although the differences are smaller, i.e., LLM-generated samples show lower mean uncertainty than SMOTE-generated samples in 33/60 (kNN, GPT), 31/60 (kNN, DEV) datasets. For DT, most points are located in the top right area showing higher uncertainty with both LLMs and SMOTE, providing not reliable results.Overall, these results support the conclusion that LLMs generate informative samples for learning the decision boundary. Some results may not be strong, therefore, we conduct an ablation study in Section 7 to support the LLM quality analyses with extensive experimental settings.

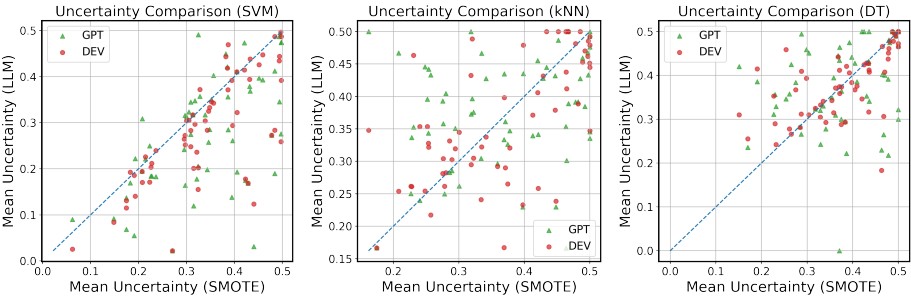

Figure 4: Uncertainty comparison between SMOTE and LLMs across multiple classifiers. Most classifiers show that LLM-generated samples has lower mean uncertainty (i.e., 42/60 (SVM, GPT), 46/60 (SVM, DEV), 33/60 (kNN, GPT), 31/60 (kNN, DEV) in datasets), respectively. DT exhibits unreliable (higher uncertainty with both LLMs and SMOTE) results. Overall results indicate LLM-generated samples provide lower uncertainty, meaning they are more informative for learning the decision boundary.

# 4 Experimental Evaluation

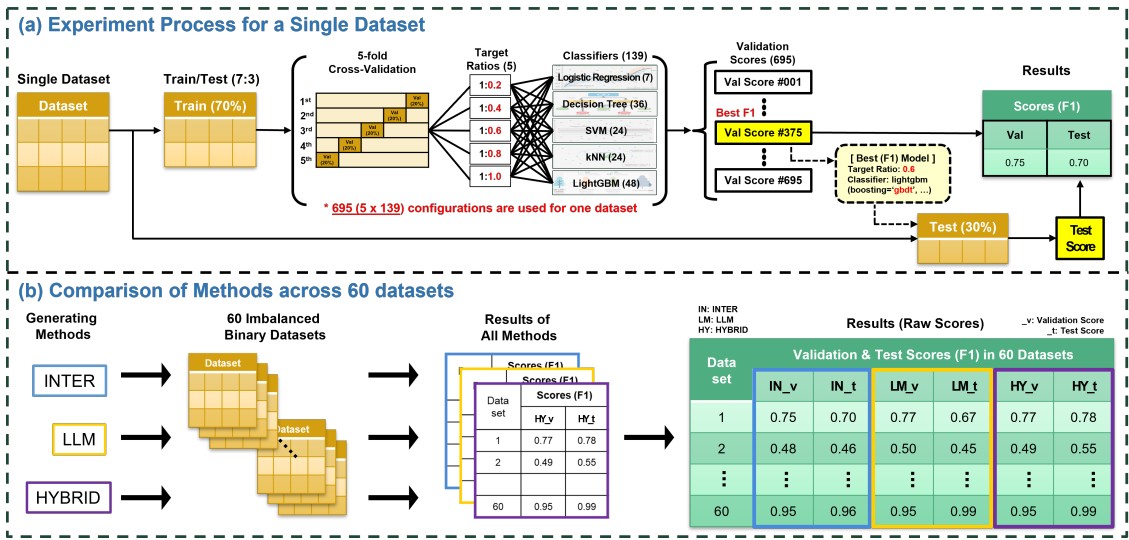

Figure 5: Experimental settings. For each method, part (a) shows the process of obtaining the final scores on a single dataset. The 5-fold cross-validation on the training data yields the best model configuration (beige dashed rectangle), which is used for testing. Part (b) shows that the goal of the experiment is to compare different data generation methods across 60 datasets. Each method goes through the process (a) with 60 datasets, resulting in the final result table (green table).

In this section, we present experimental settings followed by four research questions. The experimental settings are described in Figure 5. Part (a) shows how we obtain scores for a method in a single dataset. We employed a 5-fold cross-validation protocol, using 70% of the data for training and holding out the remaining 30% for testing. Cross-validation identifies the best-performing configuration (i.e., target ratio and classifier), and the test result is obtained using that configuration. F1 score, a standard evaluation metric for imbalanced data, is used as the primary metric. Regarding the target ratio, we set $\mathbf{A} = \{0.2, 0.4, 0.6, 0.8, 1.0\}$. For the hybrid approach, we set the seed ratio as the first target ratio (S1, $r_{seed}$ = the first $r_{target} \in \mathbf{R}_{data}$), based on the seed-ratio sensitivity analysis is provided in Section 6. Logistic Regression (LR), DT, SVM, kNN, and LightGBM (LG) are used as classifiers, yielding a total of 139 hyperparameter combinations. Details about the hyperparameters for the classifiers are provided in the Appendix A.3 (Table 4). All considered configurations are at most 695 (5 target ratios × 139 classifiers). Part (b) shows that the goal

of the experiment is to compare different methods across 60 imbalanced binary tabular datasets (Moniz & Cerqueira (2021), Table 9 in the Appendix A.7). Each method goes through the process (a) with 60 datasets, resulting in the final result table (green in the Figure). This design enables a fair comparison of different generation methods, as each method may have an optimal combination of classifier, hyperparameters, and target ratio for each dataset. The 5-fold cross-validation also ensures robust selection, yielding reliable test results. Finally, many results across 60 benchmark datasets support the practical generality of the results.

**Research Questions**  We formulate four research questions, designed to capture the central themes of our work. Each research question is directly linked to a corresponding set of experiments and analyses.

- RQ1. When interpolators are inapplicable (i.e., few-shot and zero-shot), does the hybrid approach achieve LLM-comparable performance with greater efficiency than the LLM-only method? (5.1)

- RQ2. How much does the hybrid approach provide the efficiency compared to the corresponding LLM-only methods? (5.2)

- RQ3. Does varying imbalance affect the performance of the hybrid approach? (5.3)

- RQ4. Does the hybrid approach provide the validation-to-test robustness? (5.4)

## 5    Results and Analysis

We present performance evaluations across 60 datasets using the following methods: SMOTE, Borderline SMOTE (BSM), ADASYN (ADA), LLMs (GPT and DEV), and the proposed hybrid approach (HYB). Before we address the research questions, we note that no single data generation method can dominate across datasets, which has already been discovered with prior findings in the imbalance literature (Moniz & Monteiro (2021)). We may argue that one performs better than the others if we consider a few datasets; however, results across many datasets cannot support such an argument, because the performance of a single method on a specific dataset depends on many elements, such as the dataset's characteristics, the classifier, the target ratio, etc. Under our thorough experimental settings (as practitioners do), each method will likely achieve optimal performance on a given dataset. Our experimental results also confirm the statement above, and for your reference, we present them in Appendix A.4 (Table 6).

### 5.1    Few-Shot and Zero-Shot Experiments (RQ1)

To investigate the effectiveness of the hybrid approach under extreme imbalance, we considered conditions in which the minority class constitutes only a tiny fraction of the data, i.e., few-shot and zero-shot. Specifically, we modified datasets to simulate scenarios with class ratios (major:minor) of 1:0.01 (few-shot) and 1:0.00 (zero-shot) in the training sets (the removed minority samples are moved to the validation and test sets), where standard interpolators are not applicable due to the scarcity or absence of reliable minority samples. These harsh conditions reflect the real-world challenges. To avoid biased results, we selected four diverse datasets from the work (Moniz & Cerqueira (2021)). They have different dataset sizes (S) and numbers of numerical (N) and categorical (C) features, and they cover diverse data domains (D) as shown in Table 1.

In these settings, interpolation-based methods (i.e., SMOTE, BSM, and ADA) cannot generate synthetic samples because there are insufficient minority data points. However, both LLM and HYB remained functional in these scenarios. In both few-shot and zero-shot settings, with any LLM (GPT or DEV), the LLM-only method and HYB consistently perform comparably. In a zero-shot setting, especially when standard learning without data generation (ORG) is not possible due to the absence of minority samples, we observe similar results. Importantly, there are significant runtime gaps between the LLM-only and the hybrid approaches. In all cases, HYB requires considerably less runtime than the corresponding LLM-only method, because HYB invokes the LLM only for the initial seed generation. The results highlight the practical advantage of our method in extremely scarce scenarios because HYB remains functional in the harsh conditions and achieves competitive performance with LLM, even with greater efficiency. The data visualizations of these datasets are provided in the Appendix A.8.

Table 1: [RQ1] Evaluation under extreme conditions where interpolators (SMOTE/BSM/ADA) are not applicable. In any setting (few and zero-shot) with any LLM (GPT and DEV), HYB achieves LLM-comparable performance with greater efficiency (shorter runtime).

| Comparison | | Few-Shot (1:0.01) | | | | | Zero-Shot (1:0.00) | | | | |
|---|---|---|---|---|---|---|---|---|---|---|---|
| Dataset | Eval. | Base | GPT-4o-mini | | Devstral 2 | | Base | GPT-4o-mini | | Devstral 2 | |
| {S, N, C, D} | Metric | ORG | LLM | HYB | LLM | HYB | ORG | LLM | HYB | LLM | HYB |
| Birthday (Bischl et al. (2025)) | F1 | 0.2703 | 0.6970 | **0.7246** | 0.7671 | **0.7692** | N/A | **0.4342** | **0.4342** | **0.4342** | **0.4342** |
| {335, 1, 2, Demography} | Second | N/A | 551 | **65** | 415 | **38** | N/A | 579 | **60** | 347 | **39** |
| Solar Flare (Bischl et al. (2025)) | F1 | 0.0161 | 0.5477 | **0.5726** | **0.5820** | 0.5809 | N/A | 0.1453 | **0.3599** | **0.5302** | 0.5238 |
| {1,066, 0, 7, Physics} | Second. | N/A | 3,748 | **291** | 1,730 | **143** | N/A | 2,211 | **235** | 1,900 | **144** |
| Draft (Bischl et al. (2025)) | F1 | 0.0909 | 0.2837 | **0.2857** | 0.3590 | **0.4091** | N/A | **0.3220** | 0.3182 | **0.2958** | 0.2937 |
| {365, 3, 2, Sports} | Second. | N/A | 1,267 | **127** | 678 | **51** | N/A | 1,188 | **102** | 586 | **51** |
| Blood Transfusion (Yeh et al. (2009)) | F1 | 0.0488 | **0.6053** | 0.5819 | **0.5863** | 0.5819 | N/A | 0.5540 | **0.5674** | **0.5831** | 0.4651 |
| {748, 4, 0, Medical} | Second. | N/A | 1,857 | **144** | 1,051 | **81** | N/A | 1,660 | **150** | 932 | **75** |

*S/N/C/D: #Size/#Numerical Feat./#Categorical Feat./Domain       **Bold**: Best performance or efficiency

## 5.2 Efficiency (RQ2)

While LLM-based methods generate contextually meaningful samples, they are prohibitively slow because LLMs are invoked repeatedly for each target ratio. In contrast, the hybrid approach requires the LLM only during the initial seed-generation step, after which SMOTE handles subsequent resampling at negligible computational cost. This design ensures that the hybrid approach avoids its runtime bottlenecks. To quantify this, we measured the runtime of methods on the two datasets, 'Dataset #3' and 'Dataset #13', from our 60 datasets (see Table 9). Figure 6 provides runtime comparison results between LLM and multiple hybrid approaches ($r_{\text{seed}}$: 0.2, 0.4, 0.6, 0.8) using two LLMs (GPT and DEV) on the two datasets.

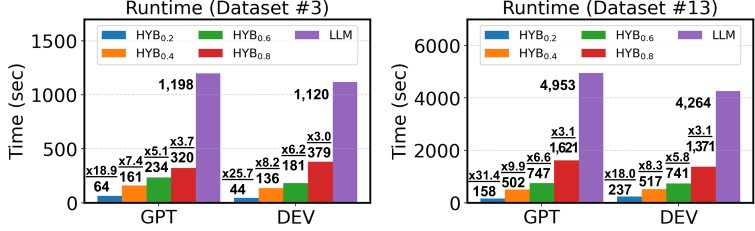

Figure 6: [RQ2] Runtime of LLM and the hybrid approach on two datasets. Across all results, any HYB takes significantly less runtime than the LLM-only method. $\text{HYB}_{\text{seed ratio: 0.2}}$ (blue) is faster 18.0 - 31.4 times than the LLM-only method (purple).

The left shows the runtimes on Dataset #3, and the right shows them on Dataset #13. In each figure, the five left bars represent the results with GPT, and the five right bars show DEV. Each method must generate multiple augmented datasets with different target ratios to find the best; therefore, for each method, we report the total runtime for generating all datasets. For example, the left-most bar (blue, Dataset #3, GPT, $\text{HYB}_{0.2}$) shows the total runtime for generating $\text{HYB}_{\text{seed: 0.2,target: \{0.4, 0.6, 0.8, 1.0\}}}$ using GPT for the Dataset #3. Specifically, $\text{HYB}_{0.2, 0.4}$ is generated by GPT until the major-to-minor ratio reaches 1:0.2, and by SMOTE until the ratio reaches 1:0.4 in 64 seconds. Thereafter, $\text{HYB}_{0.2,0.6}$ is generated in 0.0025, $\text{HYB}_{0.2,0.8}$ is generated in 0.0027, and $\text{HYB}_{0.2,1.0}$ is generated in 0.0024, since LLM-generated data points (for the ratio of 1:0.2) were reused. So the $\text{HYB}_{\text{seed ratio:0.2}}$ spent 64 seconds in total to generate four augmented datasets (i.e., $\text{HYB}_{0.2,0.4}$, $\text{HYB}_{0.2,0.6}$, $\text{HYB}_{0.2,0.8}$, and $\text{HYB}_{0.2,1.0}$). In all four results (i.e., with GPT and DEV at the Dataset #3 and #13), the LLM-only method (purple) takes significantly more runtime than any HYBs. If we use the first target ratio strategy (S1), $\text{HYB}_{\text{seed ratio: 0.2}}$ is 18.0 - 31.4 times faster than the LLM-only method. Even the slowest one, $\text{HYB}_{\text{seed ratio: 0.8}}$, is 3.0 - 3.7 times faster. The efficiency of HYB stems from its hybrid mechanism, i.e., reduced reliance on LLMs; therefore, across any LLMs and datasets, the hybrid approach is always more efficient than the LLM-only method. These results demonstrate that

the LLM-only method scales poorly, making it impractical for large-scale use, while the hybrid approach maintains constant runtime after the initial seed generation, improving efficiency. Measured runtimes of all methods by each $r_{target}$ are provided in the Appendix A.5 (Table 7, Figure 12).

## 5.3 Performance by Imbalance Severity (RQ3)

To evaluate the hybrid approach under realistic varying data imbalance, we conducted experiments on 60 imbalanced benchmark datasets (see Table 9) with diverse imbalance ratios (IR $= \frac{N_{maj}}{N_{min}}$; higher IRs indicate greater imbalance). The IRs range from considerably skewed distributions ($\sim$12) to near-balanced settings ($\sim$1.5), enabling a comprehensive assessment across different degrees of class rarity. To understand how methods behave under different IRs, we divide the 60 datasets into four groups (of a similar size) based on their IRs: IR$<$2 (16 datasets), IR$\geq$2 (14 datasets), IR$\geq$3 (15 datasets), IR$\geq$6 (15 datasets). This grouping allows us to capture trends in performance as class imbalance becomes more or less severe. To measure the effectiveness of the hybrid approach over other methods, we defined the relative improvement as in Eq. (8). The relative improvement indicates how much better the hybrid approach performs than a baseline method.

$$\Delta_{\text{rel}} baseline \ (\%) = \frac{F1_{\text{HYB}} - F1_{\text{baseline}}}{F1_{\text{baseline}}} \times 100\% \tag{8}$$

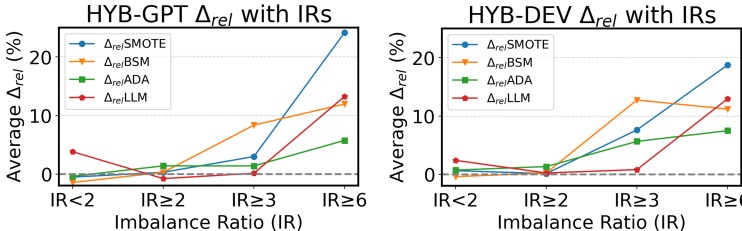

Figure 7: [RQ3] Average relative improvement of the hybrid approach (left: GPT, right: DEV) in the four IR groups. In both results, HYB shows a larger relative improvement over all other methods in more imbalanced groups. The results show that the hybrid approach becomes more effective in more imbalanced settings.

Figure 7 shows the average relative improvement in the different IR groups. Across both results (GPT and DEV), the hybrid approach shows a larger relative improvement over all other methods in higher IR groups, whereas differences are smaller in lower IR groups. For instance, HYB-GPT achieves over 10% relative improvement over SMOTE, BSM, and GPT-only in the most imbalanced group (i.e., IR$\geq$6). In the other three groups, HYB-GPT shows less than 10% relative improvement over all methods. Although some results show a slight decrease as IR increases, the overall trend indicates a positive relationship between relative improvement and IRs. In the highly imbalanced setting, interpolators are limited by their reliance on existing samples, while LLMs can expand the observed minority support and generate informative samples (see Section 3.2).

In addition, we observe that the LLM-only method performs slightly worse than interpolators in the least imbalanced group (i.e., IR$<$2), indicating interpolators are enough to represent minority distribution in near-balanced settings. In contrast, based on the observations in Section 3.2.1, while LLMs can generate potentially informative samples, they may also produce samples that lie far from the observed minority support, leading to increased variability in their usefulness. In the moderate imbalanced group (i.e., IR$\geq$2 and IR$\geq$3), the LLM-only method performs almost same as the hybrid method, i.e., it has more performance gain over interpolators as IR increases. However, in the most imbalanced group (i.e., IR$\geq$6), the LLM-only method does not outperform interpolators. This shows that the potention risk that using the LLM-only method for entire data generation may contain some noisy samples. In contrast, the hybrid approach reduces it by limiting the LLM use and leverages interpolation for more consistent data generation. This operational design enables the hybrid approach to obtain the LLMs' exploration while mitigating the risk of over spread or less informative samples. As a result, the hybrid approach achieves a more stable and effective augmentation, particularly in scenarios where the minority distribution is sparse.

## 5.4  Robustness (RQ4)

To assess robustness (i.e., validation-to-test consistency), we define the Achievement Rate (AR) as shown in Eq. (9), which captures how well a method generalizes beyond the validation set. $AR \approx 1$ indicates consistent generalization, while $AR \ll 1$ and $AR \gg 1$ implies overfitting and underfitting, respectively.

$$AR_{method} = \frac{F1_{\text{test}}}{F1_{\text{validation}}} of\ the\ method \tag{9}$$

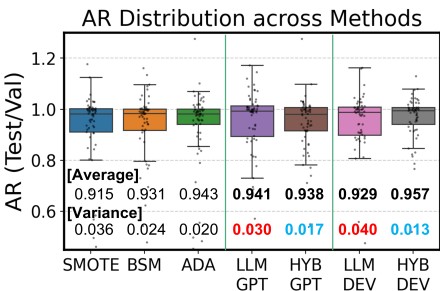

Figure 8: [RQ4] AR distribution by methods. HYB-GPT provides a similar average AR over LLM-GPT ($0.941 \rightarrow 0.938$) and a prominent reduction in variance ($0.030 \rightarrow 0.017$). HYB-DEV shows significant improvements in both (average: $0.929 \rightarrow 0.957$, and variance :$0.040 \rightarrow 0.013$) over LLM-DEV. These observations demonstrate that the hybrid method can improve robustness, as supported by its theoretical mechanism.

Fig. 8 shows the measured AR of each method across the 60 datasets in box plots. The left three are interpolators, the next two are LLM and the hybrid approach with GPT, and the last two are them with DEV. The average ARs range from the farthest from 1.0 (SMOTE: 0.915) to the closest to 1.0 (HYB-DEV: 0.957). The variance shows the consistency of AR scores across datasets, ranging from the highest (LLM-DEV: 0.040) to the lowest (HYB-DEV: 0.013). A low variance indicates consistent AR values across datasets, whereas a high variance indicates inconsistent AR values. Importantly, adopting the hybrid approach (HYB) yields significant improvements in robustness compared to the LLM-only method. Regarding GPT, HYB provides a similar average AR ($0.941 \rightarrow 0.938$) but a prominent reduction in variance ($0.030 \rightarrow 0.017$). With DEV, HYB shows significant improvements in both the average ($0.929 \rightarrow 0.957$) and the variance ($0.040 \rightarrow 0.013$). In particular, we note that the two LLMs have the highest variances (red), whereas the hybrid approach with them have the lowest (blue). This suggests the potential instability of expansion driven solely by LLMs (see Section 3.2.1). Again, this demonstrates the validity of the hybrid mechanism: the reduced LLM sufficiently provides representational depth, and subsequent generations are safely executed by interpolation within the augmented area. These observations prove that the hybrid approach can improve robustness, rooted in its operational mechanism.

## 6  Seed-ratio Sensitivity Analysis

We previously mentioned in Section 3.1.2, that a smaller seeding strategy is more efficient and yields more candidate datasets, but also is has a possible efficiency-performance trade-off. To investigate this, we conduct a sensitivity analysis that examines how the initial seed ratio affects the hybrid approach's performance. We evaluated four seeding strategies: generating LLM-based samples up to the first target ratio (S1), the second (S2), the third (S3), and the fourth (S4) target ratio, followed by interpolation to the higher target ratios. Because the major-to-minor ratios vary across datasets, we split the datasets into 3 groups based on their ratios: Group1 (G1): '<1:0.2', Group2 (G2): '<1:0.4', and Group3 (G3): '<1:0.6'. For G1, S1 indicates that $r_{\text{seed}} = 0.2$ because the minority class ratio of the datasets is less than 0.2. For G2, S1 uses 0.4 as $r_{\text{seed}}$, and 0.6 for S1 in G3. Therefore, in G1, datasets can use four hybrid approaches (i.e., S1[$r_{\text{seed}}$=0.2] - S4[$r_{\text{seed}}$=0.8]); in G2, three (i.e., S1[$r_{\text{seed}}$=0.4] - S3[$r_{\text{seed}}$=0.8]); and in G3, two (i.e., S1[$r_{\text{seed}}$=0.6] and S2[$r_{\text{seed}}$=0.8]). We compare the strategies within each group.

From the results (provided in the Appendix A.6, Table 8), although S1 achieves the highest average F1 score across all groups and the average scores within each group show that earlier seeding mostly yielded higher performance, some differences are too small, and the standard deviations are large because we are handling multiple heterogeneous datasets. Average scores may not accurately reflect the overall trend. To address this, we adopted the Bayesian Sign Test (BST) (Benavoli et al. (2017)), which estimates the probability that one method statistically outperforms another across datasets. Specifically, BST provides probabilities

of winning, drawing, and losing for probabilistic comparisons, enabling robust insights across heterogeneous datasets. BST requires a hyperparameter, the Region of Practical Equivalence (ROPE), that determines whether differences between the two candidates are considered the same. If the difference between the two exceeds ROPE, BST acknowledges that they are different. We use 0.001 as the ROPE value to capture even a trivial effect from different seeding strategies. To investigate our observation, "earlier seeding strategy yields higher performance", we compare each strategy against the others: S1 vs S2, S2 vs S3, and S3 vs S4. Figure 9 shows BST results of comparisons in each group (left: GPT and right: DEV).

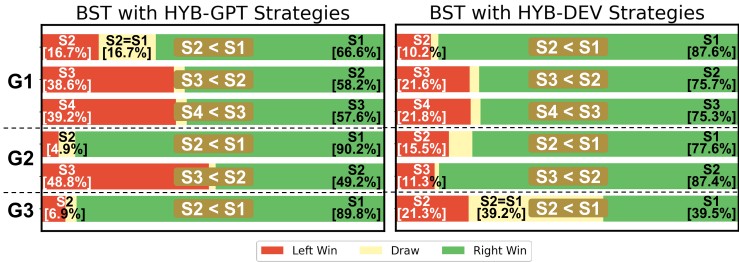

Figure 9: BST results of different strategies. Earlier seeding strategies have a higher winning probability, i.e., statistically perform better (G1: S1>S2>S3>S4, G2: S1>S2>S3, G3: S1>S2).

In the results with GPT (left), the comparison of S1 and S2 in G1 shows: {S1 > S2: 66.6%, S1 = S2: 16.7%, S1 < S2: 16.7%}. The highest probability concludes that S1 > S2. In this way, we interpret the following results: S2 has a 58.2% winning probability over S3, and S3 has a 57.6% winning probability over S4, resulting in S1 > S2 > S3 > S4 in G1. In other groups, we observe the same results: S1 > S2 > S3 in G2 and S1 > S2 in G3. The results with DEV (right) show the same pattern. Some comparisons provide competitive results; however, importantly, generating a minimal number of LLM seeds (i.e., the first-target-ratio strategy, S1) is sufficient to achieve competitive performance, and increasing LLM usage does not consistently yield better results. This suggests that the (most efficient) first target ratio strategy for the hybrid approach yields competitive performance within a reasonable range.

The sensitivity results indicate that a smaller seed is more beneficial (i.e., more efficient and sufficiently effective); therefore, it is reasonable to use a smaller seed for the hybrid approach.

# 7 Ablation Study

In Section 3.2, we provided quantifiable demonstration (i.e., analyses of LLM-seeds qualities using generated data samples) to justify LLM use as a initial seed generator. In this section, we conduct an ablation study to support the justification by isolating the contribution of LLM-based seed generation. We compare the proposed hybrid approach (LLM-seed + SMOTE) with an alternative method that uses SMOTE-generated seeds (SM+SM). Since this method is not applicable in few-shot and zero-shot settings, we conduct experiments with the 60 datasets. The results is provided in Figure 10. It shows similar pattern to other baselines as observed in Section 5.3, meaning the hybrid approach obtains larger relative performance gains as the imbalance becomes more severe. The hybrid method with GPT shows little difference in low IR groups but shows large difference in the largest IR group. With DEV, the hybrid approach shows clearer increasing pattern. LLM-seeds brought better performance than SMOTE-seeds (i.e., not expanding, interpolated seeds), which supports our previous analysis that LLMs generate informative seeds to expand the minority support. This result indicates that the initial seed samples affect the final performance, demonstrating the effectiveness of LLM-seeds.

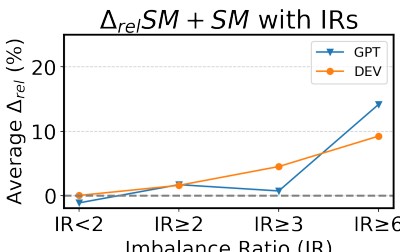

Figure 10: Average relative improvement of the hybrid approach over 'SMOTE-seeds + SMOTE (SM+SM)' method. The hybrid approach has more performance margin as IR increases.

## 8 Discussion

In this section, we discuss the details about its practical usage about the hybrid approach and address the limitations. Table 2 summarizes the research questions with corresponding experimental results.

Table 2: Comparisons of all methods using research questions.

| Research Question | LLM | Interpolators | Ours | Comment | Reference |
|---|---|---|---|---|---|
| (RQ1) Functionality in extreme imbalance | ✔ | N/A | ✔ | Ours achieves LLM-comparability efficiently | Table 1 |
| (RQ2) Efficiency (Scalability) | ✘ | ✔ | ▲ | Ours enables significant runtime reduction | Figure 6 |
| (RQ3) Ours performance by IR severity | ▲ | ▲ | ✔ | More performance gains on Ours as IR increases | Figure 7 |
| (RQ4) Robustness (Stability) | ✘ | ▲ | ✔ | Ours provides lower AR variance over LLM-only | Figure 8 |

✘: Limited, ▲: Moderate, ✔: Best

Discussion 1: When is the hybrid approach useful? Answer: The hybrid approach is effective under severe imbalanced settings, i.e., few-shot, zero-shot, and highly imbalanced (e.g., IR≥6 in this paper). The specific degree of imbalance cannot be answered, but its effectiveness under these imbalanced settings are examined with the experiments.

Discussion 2: Is the efficiency (runtime or scalable) of the hybrid approach valid with any combinations of LLMs and interpolators? Answer: Yes. The efficiency stems on the hybrid design constraining the LLM use (i.e., LLMs are used only for a small number of seeds generation); therefore, its efficiency over the corresponding LLM-only method is valid with any hybrid combinations.

Limitation 1: As shown in Section 3.2, not all LLM-seeds are helpful for the downstream task (i.e., some of them are too far from the original data support or not informative). In this work, we did not filter them out. If we improve the quality of the LLM-seeds by selecting better ones, this can contribute to improving the effectiveness of the hybrid approach. Filtering LLM-seeds is worth for the future research.

Limitation 2: The only bottleneck of the hybrid approach is the generating seeds part since LLMs require heavier computational cost than interpolation. Which means that a smaller seed ratio (i.e., a fewer number of LLM-seeds) can provide better efficiency. Although we suggest the first target ratio strategy for the seed ratio choice, there still is room for improvement, i.e., for a specific dataset and classifiers, this strategy may not be optimal. We can consider a meta learner using the meta features of a dataset (e.g., sample number, feature number, outlier ratio, etc.) to choose the seed ratio, which will have a great impact to the usage of the hybrid approach.

## 9 Conclusion

This paper introduces the hybrid synthetic tabular data generation approach that leverages the complementary strengths of LLMs and interpolators to address imbalanced tabular data. The analyses of the seed sample quality provide justifies the LLM-seeds, and extensive experimental results verify its effectiveness. As a foundational work of this new hybrid concept, we provide baseline results across 60 imbalanced benchmark datasets for future research. We discuss the details and limitations of the hybrid approach, providing both practical and research insights. For reproducibility, all the code and datasets for experiments in this work are available at `https://anonymous.4open.science/r/LSH-D711/README.md`.

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

## A  Appendix

### A.1  Our LLM-based Method

# LLM-Based Tabular Data Generator

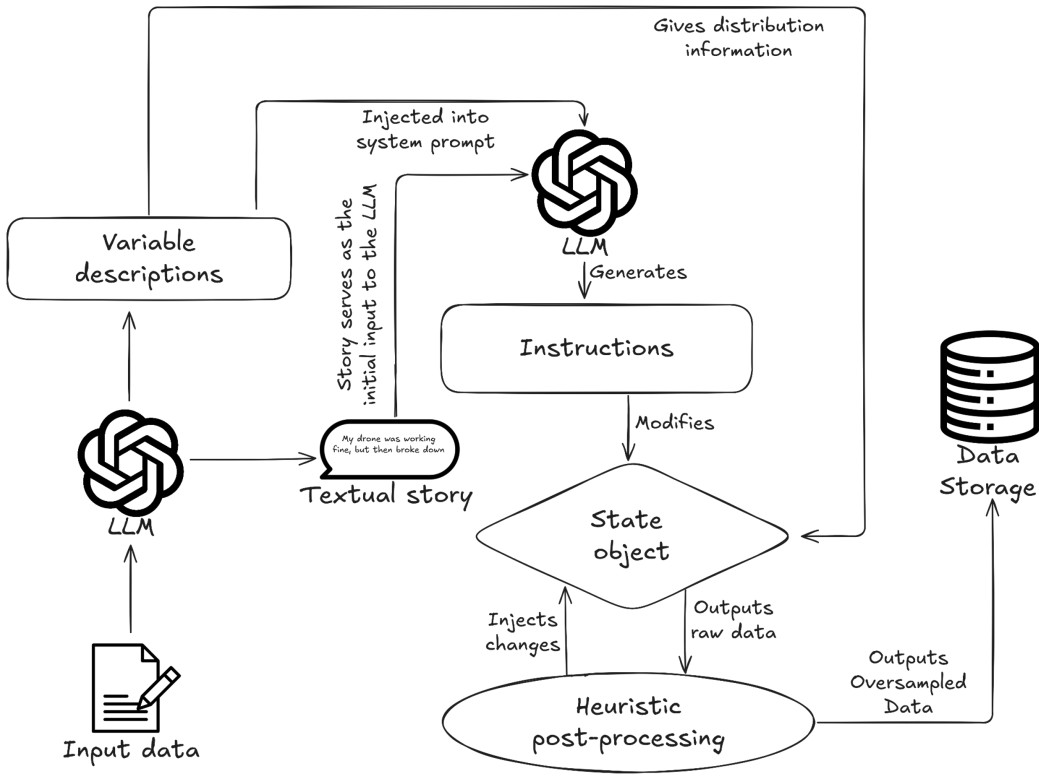

Figure 11: The mechanism of our LLM-based method, which uses the input tabular data, LLM, and prompts.

The data generation pipeline proceeds in several stages as described in Figure 11. First, we draw a sample from the dataset we are trying to augment and prompt the LLM to generate a rich textual description of it. Additionally, we use constrained generation to extract descriptions and potential distributions of the dataset's variables from the provided sample.

Next, the LLM is prompted to generate a sequence of instructions for an abstract machine. In this procedure, the instruction set is provided to the LLM, and constrained generation is used to ensure syntactically correct code. The instruction set includes instructions for setting a variable to a value, incrementing a variable by a delta, and outputting the current state of the machine. On output, the machine's outputs are jittered by a small amount, utilizing the distributions extracted by the model. The outputs of the abstract machine constitute the augmented data. We then rerun this pipeline until we obtain the desired number of synthetic samples.

This method maximizes the use of current LLMs' code-generation capabilities while avoiding the direct generation of large amounts of numerical data. At the same time, this method does not require access to model logits or any other outputs beyond the generated tokens; this allows it to be used with both open-source models and LLMs available only via limited APIs.

Furthermore, we provide the LLM prompts we used on each step of the data generation pipeline described in Figure 11.

**[Data description prompt]:**

You're an expert data analyst at a large company. Your boss has asked you to
    analyze a dataset and provide a report on your findings.

You'll be provided with a dataset in CSV format with a header. You'll need to
    write a report that provides insights into the data. Focus on the key
    points that will help your boss make informed decisions. Put a scpecial
    emphasis on the relationships between the different variables in the
    dataset, the trends that you observe.

Make sure to try to extract the general meaning of the data, and not just the
    raw numbers. Your boss is looking for a high−level summary of the data,
    not a detailed analysis.

Your report should be clear, concise, and easy to understand. It should be
    written in plain English, and should not contain any technical jargon. At
    least 500 words.

**[Variable description prompt]:**

You're an expert data analyst at a large company. Your boss has asked you to
    analyze a dataset and provide a report on your findings.

You'll be provided with a dataset in CSV format with a header. Your task is to
     extract the names, descriptions and possible distributions of the
    variables in the dataset. Please note that categorical and string
    variables use the "none" distribution.

You should provide your answer as a machine−readable JSON object. The object
    should follow this schema:

$VARIABLE−EXTRACTION−JSON−SCHEMA

* Where `$VARIABLE-EXTRACTION-JSON-SCHEMA` is a dynamically generated JSON schema of the desired
output format (described in code as a Pydantic model) that is also enforced by constrained generation.

**[Data generation prompt]:**

Your task is to control a synthetic data generator that creates synthetic data
     conforming to a report given to you.

The generator is a state machine that has an internal state which can be sent
    to the output, or that can be changed. You can use functions to control
    the state machine. Each output of the machine describes a single data
    point.

You will be given a report by the user that the synthetic data should follow
    as closely as possible. Ensure that you only use the variables provided to
     you by the user. Make sure that the synthetic data you generate is
    consistent with the report. Make sure to use the EXACT variable names
    provided in the report.

You should just output the sequence of operations without any text in natural
    language.

The change_by operation cannot be used on variables with units "true or false"
    or string variables.

Your output should be a single JSON object following the schema below:
$DATA–GENERATION–JSON–SCHEMA

Do not use the 'change_by' operation on variables with units "true or false",
    string or categorical variables.

The sequence of operations should output at least $N–OUT data points.

An example of the data that you should try to generate is:
```
$DATA–EXAMPLE
```

(this is just an example of the data format, do not try to replicate the data
    itself, only the overall format)

The sequence of operations should be consistent with the report and follow the
    schema above.

You must generate a sequence of instructions that will output at least $N–OUT
    data points.

You have access to the following variables (their exact names between |
    symbols):
$VARIABLE–DESCRIPTION–TEXT

* Where `$DATA-GENERATION-JSON-SCHEMA` is a dynamically generated JSON schema describing the output format (also described as a Pydantic model, and enforced via constrained generation); `$N-OUT` is the desired count of generated samples (provided by the user); `$DATA-EXAMPLE` is a small sample of the pipeline input data; `$VARIABLE-DESCRIPTION-TEXT` is a dynamically generated textual version of the variable descriptions (names, units, minimum and maximum values, distributions) as extracted by the model.

## A.2 Data Generation Example

We provide the details of the application of the hybrid approach. Let $A = \{r_i | 0 < r_i < r_{i+1} \leq 1.0, \ i \in \mathbb{Z}\}$ be a set of all target ratios and $R_{data} = \{r_i \in A | \frac{N_{min,data}}{N_{maj,data}} < r_i\}$ be a set of target ratios available for a specific dataset, which is determined by the minority class ratio of the dataset (i.e., $\frac{N_{min,data}}{N_{maj,data}}$). Then, a seed ratio for the LLM can be chosen, where $r_{seed} \in \mathbf{R}_{data} - max\{\mathbf{R}_{data}\}$ and $r_{target} \in \{r_i \in R_{data} | r_{seed} < r_i \}$.

Consider a dataset ($N_{maj} = 1,000$, $N_{min} = 300$) and $\mathbf{A} = \{0.2, 0.4, 0.6, 0.8, 1.0\}$. Then, $\mathbf{R}_{data}$ is determined as $\{0.4, 0.6, 0.8, 1.0\}$, because the minority class ratio ($0.3 = \frac{300}{1,000}$) is smaller than 0.4. The possible seed ratios, $r_{seed}$, are 0.4, 0.6, and 0.8, since the $max\{\mathbf{R}_{data}\}$ (i.e., 1.0) is removed. Each LLM- and interpolator-only method simply generates four augmented datasets with the four target ratios ($\in \mathbf{R}_{data}$), and the best-performing is selected for each method (e.g., $LLM_{0.6}$ and $INTER_{0.8}$).

In contrast, the hybrid approach can have multiple strategies depending on the seed ratio. In the example, there exist three strategies ($r_{seed} = 0.4, 0.6,$ or $0.8$). With the seed ratio of 0.4 (the first $r_{target}$), three target ratios are available (i.e., $r_{seed} = 0.4$ and $r_{target} = 0.6, 0.8,$ or $1.0$). We call this strategy the first

target ratio strategy (S1). An LLM generates samples up to the first target ratio (1:0.4 in this example), and the interpolator further generates samples at target ratios 0.6, 0.8, and 1.0. Similarly, the second target ratio strategy (S2) generates two augmented datasets, i.e., $HYBRID_{0.6,0.8}$ and $HYBRID_{0.6,1.0}$, where $HYBRID_{r_{seed},r_{target}}$. Note that target ratios can vary across datasets; for example, if a dataset's original minority ratio is 0.1, the first target ratio is 0.2. Table 3 shows the example above, i.e., a dataset with $N_{maj} = 1,000$, $N_{min} = 300$ and A = {0.2, 0.4, 0.6, 0.8, 1.0}. Then, $R_{data}$ is {0.4, 0.6, 0.8, 1.0}. The possible seed ratios, $r_{seed}$, are 0.4, 0.6, or 0.8.

Table 3: Data generation details for the dataset ($N_{maj} = 1,000$, $N_{min} = 300$), A = {0.2, 0.4, 0.6, 0.8, 1.0}.

| Method | Generation by Target Ratio ($\underline{N_{\text{LLM}}}+N_{\text{INTER}}$) | | | | Total |
|---|---|---|---|---|---|
| | $r_{target}$=0.4 ($N_{\text{syn}}$=100) | $r_{target}$=0.6 ($N_{\text{syn}}$=300) | $r_{target}$=0.8 ($N_{\text{syn}}$=500) | $r_{target}$=1.0 ($N_{\text{syn}}$=700) | LLM Gen. |
| INTER | $INTER_{0.4}$ ($\underline{0}$+100) | $INTER_{0.6}$ ($\underline{0}$+300) | $INTER_{0.8}$ ($\underline{0}$+500) | $INTER_{1.0}$ ($\underline{0}$+700) | $\underline{0}$ |
| LLM | $LLM_{0.4}$ ($\underline{100}$+0) | $LLM_{0.6}$ ($\underline{300}$+0) | $LLM_{0.8}$ ($\underline{500}$+0) | $LLM_{1.0}$ ($\underline{700}$+0) | $\underline{1,600}$ |
| HYB-S1 ($r_{\text{seed}}$=0.4) | - | $HYB_{0.4,0.6}$ ($\underline{100}$+200) | $HYB_{0.4,0.8}$ ($\underline{100}$+400) | $HYB_{0.4,1.0}$ ($\underline{100}$+600) | $\underline{100}$ |
| HYB-S2 ($r_{\text{seed}}$=0.6) | - | - | $HYB_{0.6,0.8}$ ($\underline{300}$+200) | $HYB_{0.6,1.0}$ ($\underline{300}$+400) | $\underline{300}$ |
| HYB-S3 ($r_{\text{seed}}$=0.8) | - | - | - | $HYB_{0.8,1.0}$ ($\underline{500}$+200) | $\underline{500}$ |

$LLM_{r_{target}}$, $INTER_{r_{target}}$, $HYB_{r_{seed},r_{target}}$
Underline: Number of LLM-generated Data

Specifically ,for example, $HYBRID_{0.4,0.6}$ ($r_{\text{seed}} = 0.4$, $r_{\text{target}} = 0.6$) is generated as follows: an LLM generates 100 samples (the major-to-minor ratio becomes 1:0.4, $r_{\text{seed}}$), and an interpolator generates 200 more samples relying on the 400 samples (i.e., original minority 300 samples + LLM-generated 100 samples), whose ratio becomes 1:0.6 ($r_{\text{target}}$). For further target ratios, such as $HYBRID_{0.4,0.8}$ and $HYBRID_{0.4,1.0}$, the hybrid approach reuses the 100 LLM-generated samples; therefore, the total number of LLM-generated samples for the HYBRID ($r_{\text{seed}} = 0.4$) is 100. In this way, **as $r_{\textbf{seed}}$ increases** (S1: $0.4 \rightarrow$ S2: $0.6 \rightarrow$ S3: $0.8$), we observe: **1) The number of LLM-generated data increases** (S1: $100 \rightarrow$ S2: $300 \rightarrow$ S3: $500$), and **2) The number of candidates (i.e., augmented datasets) decreases** (S1: 3 ($HYBRID_{0.4,\{0.6,0.8,1.0\}}$) $\rightarrow$ S2: 2 ($HYBRID_{0.6,\{0.8,1.0\}}$) $\rightarrow$ S3: 1 ($HYBRID_{0.8,1.0}$)).

As shown in the table, each INTER and LLM generates four augmented datasets at the four target ratios, and the best-performing is selected (e.g., $LLM_{0.6}$ and $INTER_{0.8}$) based on validation performance. Although this table does not include performance information, we can still assess each method's efficiency. The last column ("Total LLM Gen.") shows the number of data samples generated by LLM. Interpolation-based methods, such as SMOTE, use simple mathematical calculations to generate data, whereas LLMs rely on highly complex, intensive computations; therefore, interpolators are more efficient than LLMs. We can compare the efficiency of different methods based on the number of LLM-generated samples. In this example, the LLM-only method requires 1,600 LLM-generated samples, which is less efficient than INTER (which requires no LLM-generated samples).

## A.3 Classifiers

Table 4 shows all hyperparameter settings for the five classifiers.

Table 4: the 139 hyperparameter settings of classifiers.

| Classifier | Combination | Hyperparameters |
|---|---|---|
| LR | 7 | C:[0.001, 0.01, 0.1, 1, 10, 100, 1000] |
| DT | 4X3X3 =36 | max depth:[10,20,30,40] min samples split:[2,4,6] min samples leaf:[1,2,3] |
| SVM | 3X2X4 =24 | C:[0.1, 1, 10] kernel:[rbf, sigmoid] gamma:[scale, auto, 0.1, 1] |
| kNN | 4X2X3 =24 | n neighbors:[3, 5, 7, 9] p:[1, 2] metric:[euclidean, manhattan, minkowski] |
| LG | 3X2X2X4 =48 | boosting type : [gbdt, dart, goss] max depth : [10,20] learning rate : [0.01,0.05] n estimators: [50, 100, 150, 200] |
| Total | 139 | |

### A.4 Overall Performance

We compare all methods across 60 imbalanced datasets. Table 5 shows the average score for all methods across the 60 datasets. The results show that none of the methods universally dominates.

Table 5: All methods' average F1 score of 60 datasets.

| Method | No Data-Generation | Interpolator INTER | BSM | ADA | GPT LLM | HYB | DEV LLM | HYB |
|---|---|---|---|---|---|---|---|---|
| mean | 0.6507 | 0.6766 | 0.6761 | 0.6864 | 0.6764 | 0.6878 | 0.6768 | **0.6901** |
| std | ±0.2859 | ±0.2656 | ±0.2566 | ±0.2531 | ±0.2622 | ±0.2457 | ±0.2651 | **±0.2430** |

**Bold**: Best, Underline: Second best

Although the hybrid approach (HYB) achieves the highest score, the differences among the methods are not significant, ranging from 0.6761 to 0.6901. We use multiple heterogeneous datasets, so the average score may not accurately reflect the trend. Across two settings (GPT and DEV), the hybrid approach achieves the highest score; therefore, we compare it with the others as shown in Table 6.

Comparisons of each method with HYB are shown. The average performance margins in the third column indicate that there are no big differences between pairs. A 1% difference in average score may not be negligible, but it is not significant when evaluating multiple heterogeneous datasets. The fourth column ("Win Number") shows the number of datasets for which one is better than tied with/worse than the other. For example, when GPT is used, HYB outperforms SMOTE on 24 datasets, ties SMOTE on 13, and underperforms on 23. In all comparisons, we do not observe significant differences between the two methods. To statistically evaluate results across multiple datasets, we employ the Bayesian Sign Test (BST) (Benavoli et al. (2017)), which provides the probability that one method outperforms another. More specifically, the test provides probabilities of winning, drawing, and losing for comparisons across heterogeneous datasets. BST results in the fourth column (i.e., BST prob.) confirm the earlier observations (i.e., no clear superiority). In most comparisons, the drawing probabilities dominate, and the differences between win and lost probabilities are not large. This shows that none of the methods dominate across many datasets.

Table 6: Performance comparison across 60 datasets.

| Used LLM | Comparison | Average Margin | Win Number (BST Prob.) |
|---|---|---|---|
| GPT-4o -mini | HYB vs SMOTE | HYB−SMOTE : 1.1% | HYB>SMOTE: 24 (18%) HYB=SMOTE: 13 (80%) HYB<SMOTE: 23 (2%) |
| | HYB vs BSM | HYB−BSM : 1.2% | HYB>BSM: 27 (11%) HYB>BSM: 13 (87%) HYB>BSM: 20 (2%) |
| | HYB vs ADA | HYB−ADA : 0.1% | HYB>ADA: 25 (9%) HYB>ADA: 8 (67%) HYB>ADA: 27 (24%) |
| | HYB vs LLM | HYB−LLM : 1.1% | HYB>LLM: 23 (8%) HYB>LLM: 14 (88%) HYB>LLM: 23 (4%) |
| Devstral 2 | HYB vs SMOTE | HYB−SMOTE : 1.3% | HYB>SMOTE: 25 (21%) HYB>SMOTE: 12 (50%) HYB>SMOTE: 23 (29%) |
| | HYB vs BSM | HYB−BSM : 1.4% | HYB>BSM: 29 (53%) HYB>BSM: 10 (41%) HYB>BSM: 21 (6%) |
| | HYB vs ADA | HYB−ADA : 0.4% | HYB>ADA: 23 (39%) HYB>ADA: 11 (22%) HYB>ADA: 26 (39%) |
| | HYB vs LLM | HYB−LLM : 1.3% | HYB>LLM: 29 (76%) HYB>LLM: 9 (8%) HYB>LLM: 22 (16%) |

## A.5 Runtime Comparison

Table 7 shows the measured runtime of all methods on the two datasets for each $r_{target}$. For each method, we can compare the total runtime. Interpolators have incomparably lower runtime than the others, and the LLM-only method has the longest runtime. HYBs are located between the two, and the HYB with a smaller seed ratio runs faster than the one with a larger seed ratio. Figure 12 shows the cumulative runtime of all methods with each $r_{target}$, which shows the visualization of the total runtime across methods.

Table 7: Runtime of all methods with two datasets.

| Used LLM | Method | Runtime for Dataset #3 (second) | | | | | | Runtime for Dataset #13 (second) | | | | | |
|---|---|---|---|---|---|---|---|---|---|---|---|---|---|
| | $r_{target}$ ($N_{syn}$) | 0.2 (19) | 0.4 (56) | 0.6 (93) | 0.8 (130) | 1.0 (168) | Total (466) | 0.2 (8) | 0.4 (40) | 0.6 (72) | 0.8 (104) | 1.0 (136) | Total (360) |
| N/A | SMOTE | 0.0033 | 0.0027 | 0.0026 | 0.0026 | 0.0025 | 0.0138 | 0.0062 | 0.0056 | 0.0055 | 0.0055 | 0.0055 | 0.0283 |
| | BSM | 0.0034 | 0.0031 | 0.0031 | 0.0032 | 0.0030 | 0.0158 | 0.0063 | 0.0058 | 0.0058 | 0.0058 | 0.0059 | 0.0296 |
| | ADA | 0.0032 | 0.0030 | 0.0030 | 0.0030 | 0.0030 | 0.0151 | - | 0.0059 | 0.0057 | 0.0059 | 0.0057 | 0.0232 |
| GPT -4o -mini | LLM | 63.49 | 161.16 | 234.05 | 320.06 | 419.54 | 1198.31 | 157.94 | 502.00 | 746.96 | 1621.38 | 1924.88 | 4953.15 |
| | HYB$_{0.2}$ | - | 63.49 | 0.0025 | 0.0027 | 0.0024 | 63.50 | - | 157.94 | 0.0053 | 0.0050 | 0.0050 | 157.96 |
| | HYB$_{0.4}$ | - | - | 161.17 | 0.0025 | 0.0029 | 161.17 | - | - | 502.01 | 0.0053 | 0.0050 | 502.02 |
| | HYB$_{0.6}$ | - | - | - | 234.05 | 0.0028 | 234.06 | - | - | - | 746.96 | 0.0054 | 746.97 |
| | HYB$_{0.8}$ | - | - | - | - | 320.07 | 320.07 | - | - | - | - | 1621.38 | 1621.38 |
| Dev- stral 2 | LLM | 43.63 | 135.87 | 181.30 | 379.07 | 379.71 | 1119.58 | 237.07 | 516.54 | 740.67 | 1371.33 | 1398.33 | 4263.95 |
| | HYB$_{0.2}$ | - | 43.64 | 0.0025 | 0.0025 | 0.0025 | 43.64 | - | 237.09 | 0.0050 | 0.0050 | 0.0051 | 237.09 |
| | HYB$_{0.4}$ | - | - | 135.87 | 0.0026 | 0.0026 | 135.88 | - | - | 516.55 | 0.0050 | 0.0050 | 516.55 |
| | HYB$_{0.6}$ | - | - | - | 181.31 | 0.0027 | 181.31 | - | - | - | 740.68 | 0.0053 | 740.68 |
| | HYB$_{0.8}$ | - | - | - | - | 379.07 | 379.07 | - | - | - | - | 1371.34 | 1371.34 |

HYB$_{r_{seed}}$

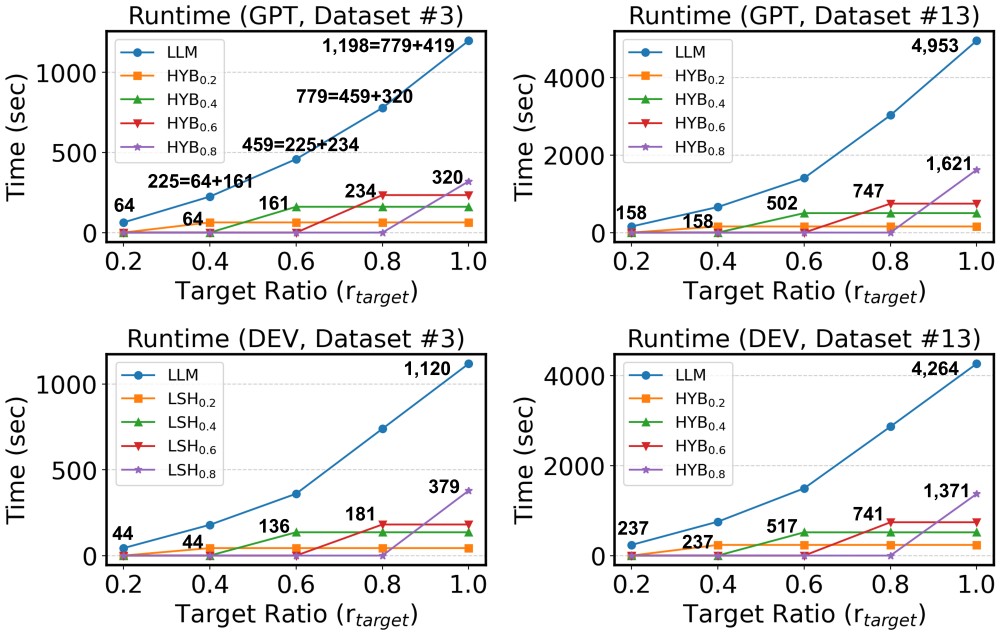

Figure 12: Cumulative runtime by target ratio. In all results, LLM requires significantly more time for data generation than any of the HYBs. As the target ratio increases, LLM's runtime increases while HYBs' remain constant.

## A.6 Seeding Strategy Analysis

Table 8 shows the averaged results of datasets by strategies in each group.

Table 8: Average F1 score by different seeding strategies in three groups. Across all groups, S1 achieves the highest average score. Best in Bold.

| Data# | Used | F1 score by Seeding Strategy (mean±std) | | | |
|---|---|---|---|---|---|
| (maj:min) | LLM | S1 | S2 | S3 | S4 |
| Group1 | | ($r_{\text{seed}}$=0.2) | ($r_{\text{seed}}$=0.4) | ($r_{\text{seed}}$=0.6) | ($r_{\text{seed}}$=0.8) |
| #1-#20 | GPT | **0.5979**±0.3008 | 0.5792±0.3291 | 0.5731±0.3157 | 0.5654±0.3408 |
| ($< 1$:0.2) | DEV | **0.6014**±0.3053 | 0.5832±0.2833 | 0.5855±0.3106 | 0.5741±0.2935 |
| Group2 | | ($r_{\text{seed}}$=0.4) | ($r_{\text{seed}}$=0.6) | ($r_{\text{seed}}$=0.8) | |
| #21-#33 | GPT | **0.6139**±0.2242 | 0.5948±0.2250 | 0.5967±0.2282 | N/A |
| ($< 1$:0.4) | DEV | **0.6199**±0.2321 | 0.6071±0.2362 | 0.6068±0.2338 | N/A |
| Group3 | | ($r_{\text{seed}}$=0.6) | ($r_{\text{seed}}$=0.8) | | |
| #34-#55 | GPT | **0.7787**±0.1851 | 0.7703±0.1910 | N/A | N/A |
| ($< 1$:0.6) | DEV | **0.7364**±0.1935 | 0.7325±0.2043 | N/A | N/A |

## A.7 Dataset Description

Table 9 describes 60 imbalanced binary tabular datasets.

Table 9: Description of 60 Imbalanced Datasets.

| Data# | Size | Feat. | IR (Maj:Min) | Data Name | Area |
|---|---|---|---|---|---|
| 1 | 403 | 35 | 12.00 (1:0.083333) | mw1 | Software Engineering |
| 2 | 661 | 37 | 11.71 (1:0.085386) | PizzaCutter1 | Software Engineering |
| 3 | 365 | 5 | 10.77 (1:0.092814) | analcatdata_draft | Sports |
| 4 | 705 | 37 | 10.56 (1:0.094720) | PieChart1 | Software Engineering |
| 5 | 990 | 13 | 10.00 (1:0.100000) | vowel | Speech/Audio |
| 6 | 504 | 19 | 9.72 (1:0.102845) | meta | Meta-Learning (Stats) |
| 7 | 458 | 38 | 9.65 (1:0.103614) | kc3 | Software Engineering |
| 8 | 1320 | 17 | 9.65 (1:0.103679) | analcatdata_halloffame | Sports |
| 9 | 2000 | 6 | 9.00 (1:0.111111) | mfeat-morphological | Image Processing |
| 10 | 1043 | 37 | 7.21 (1:0.138646) | PizzaCutter3 | Software Engineering |
| 11 | 450 | 3 | 7.18 (1:0.139241) | analcatdata_apnea3 | Medical |
| 12 | 475 | 3 | 6.79 (1:0.147343) | analcatdata_apnea1 | Medical |
| 13 | 327 | 37 | 6.79 (1:0.147368) | CastMetal1 | Industrial |
| 14 | 475 | 3 | 6.42 (1:0.155718) | analcatdata_apnea2 | Medical |
| 15 | 2310 | 17 | 6.00 (1:0.166667) | segment | Image Processing |
| 16 | 559 | 4 | 5.99 (1:0.167015) | arsenic-female-bladder | Medical |
| 17 | 470 | 13 | 5.71 (1:0.175000) | thoracic-surgery | Medical |
| 18 | 381 | 38 | 5.57 (1:0.179567) | water-treatment | Environmental |
| 19 | 500 | 22 | 5.25 (1:0.190476) | collins | Medical/Ecology |
| 20 | 562 | 21 | 5.11 (1:0.195745) | soybean | Agricultural |
| 21 | 1066 | 7 | 4.86 (1:0.205882) | solar-flare | Physics/Space |
| 22 | 797 | 4 | 4.14 (1:0.241433) | analcatdata_dmft | Medical/Dental |
| 23 | 522 | 20 | 3.88 (1:0.257831) | kc2 | Software Engineering |
| 24 | 1324 | 10 | 3.53 (1:0.282946) | mofn-3-7-10 | Synthetic/Logical |
| 25 | 1156 | 5 | 3.52 (1:0.284444) | socmob | Social Science |
| 26 | 400 | 5 | 3.44 (1:0.290323) | analcatdata_germangss | Financial |
| 27 | 812 | 6 | 3.34 (1:0.299200) | unknown | Industrial |
| 28 | 363 | 8 | 3.27 (1:0.305755) | braziltourism | Geographic/Business |
| 29 | 748 | 4 | 3.20 (1:0.312281) | blood-transfusion-service-center | Medical/Logistics |
| 30 | 336 | 14 | 3.10 (1:0.322835) | primary-tumor | Medical |
| 31 | 846 | 18 | 2.88 (1:0.347134) | vehicle | Image Processing/Automotive |
| 32 | 1000 | 20 | 2.86 (1:0.349528) | autoUniv-au1-1000 | Synthetic/Automotive |
| 33 | 306 | 3 | 2.78 (1:0.360000) | haberman | Medical |
| 34 | 583 | 10 | 2.49 (1:0.401442) | ilpd | Medical |
| 35 | 1728 | 6 | 2.34 (1:0.428099) | car | Social Science/Automotive |
| 36 | 1000 | 19 | 2.33 (1:0.428571) | credit-g | Financial |
| 37 | 358 | 31 | 2.23 (1:0.449393) | dermatology | Medical |
| 38 | 328 | 32 | 2.22 (1:0.451327) | analcatdata_marketing | Business/Marketing |
| 39 | 641 | 19 | 2.16 (1:0.463470) | eucalyptus | Ecology/Forestry |
| 40 | 593 | 77 | 2.14 (1:0.467822) | emotions | Audio/Psychology |
| 41 | 310 | 6 | 2.10 (1:0.476190) | vertebra-column | Medical |
| 42 | 2201 | 2 | 2.10 (1:0.477181) | Titanic | History/Social Science |
| 43 | 1074 | 16 | 2.09 (1:0.479339) | colleges_aaup | Education/Socioeconomic |
| 44 | 973 | 9 | 2.02 (1:0.494624) | xd6 | Synthetic/Rule-Based |
| 45 | 320 | 6 | 2.00 (1:0.502347) | pc1_req | Software Engineering |
| 46 | 1055 | 32 | 1.96 (1:0.509299) | qsar-biodeg | Chemistry/Toxicology |
| 47 | 462 | 9 | 1.89 (1:0.529801) | sa-heart | Medical |
| 48 | 958 | 9 | 1.87 (1:0.530351) | tic-tac-toe | Game/AI |
| 49 | 768 | 8 | 1.86 (1:0.536000) | diabetes | Medical |
| 50 | 683 | 9 | 1.79 (1:0.538288) | breast-w | Medical |
| 51 | 351 | 33 | 1.78 (1:0.560000) | ionosphere | Physics/Space |
| 52 | 250 | 12 | 1.77 (1:0.562500) | Horse Colic | Medical/Veterinary |
| 53 | 959 | 40 | 1.77 (1:0.564437) | tokyo1 | Image Processing |
| 54 | 609 | 7 | 1.73 (1:0.577720) | kdd_el_nino-smal | Earth Science/Climate |
| 55 | 569 | 30 | 1.68 (1:0.593838) | wdbc | Medical |
| 56 | 392 | 8 | 1.67 (1:0.600000) | cars | Automotive |
| 57 | 349 | 31 | 1.64 (1:0.608295) | cylinder-bands | Industrial |
| 58 | 250 | 9 | 1.60 (1:0.623377) | Horse Colic | Medical/Veterinary |
| 59 | 500 | 25 | 1.55 (1:0.644737) | unknown | Unknown/General |
| 60 | 4601 | 7 | 1.54 (1:0.650287) | spambase | Information Technology |

IR: Imbalance Ratio (major:minor = X:1)

## A.8 Visualization - Extreme Imbalance

Figure 13 shows the GPT-augmented dataset distributions (with PCA) of extremely imbalanced datasets. Across the four datasets, LLM more aggressively generates new synthetic samples, while HYB more gently fills the augmented area based on LLM-seeds.

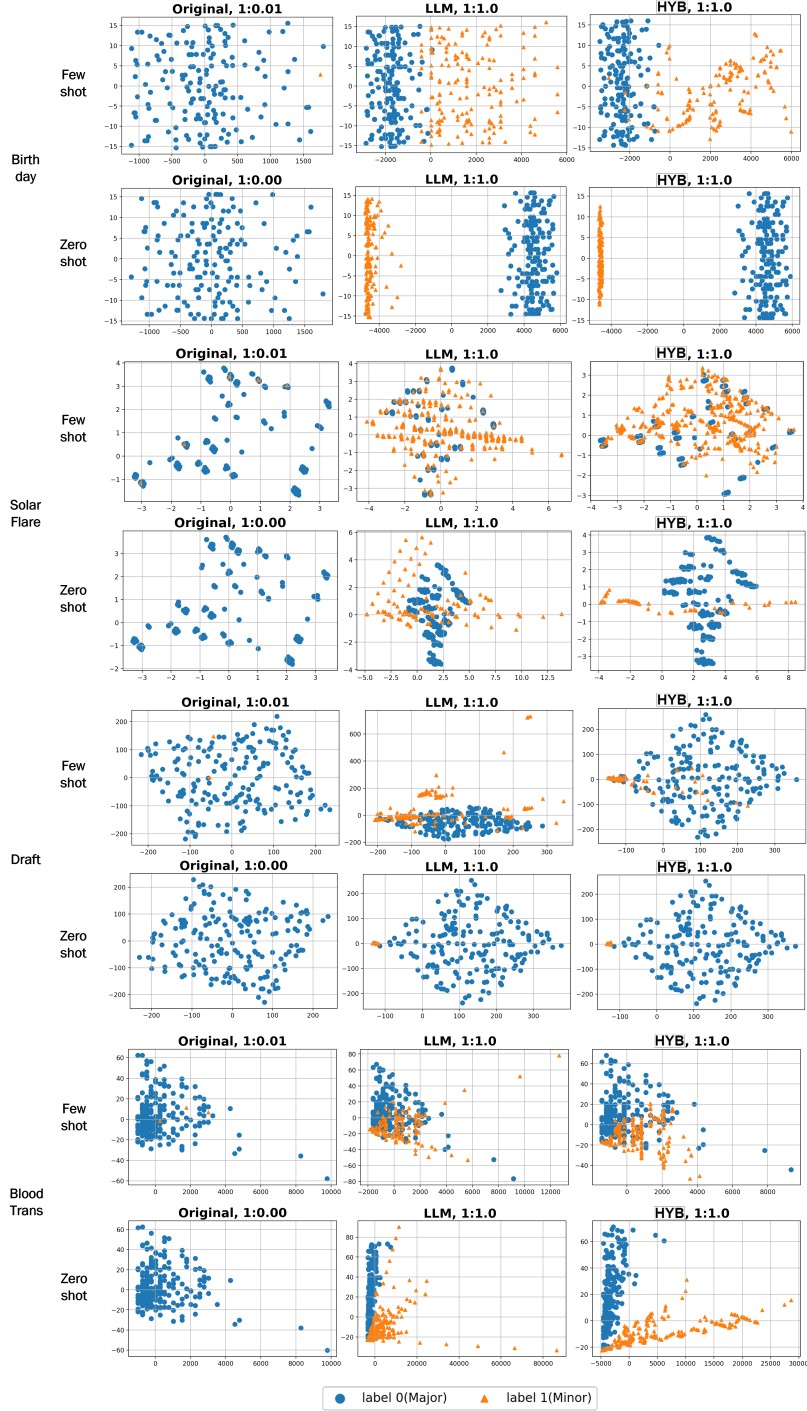

Figure 13: GPT-Visualization (with PCA) of the four datasets under few-shot/zero-shot scenarios.

