# OpenReview forum: "Imbalanced Tabular Data Synthesis via LLM-Seeds and Interpolation"
_TMLR — Under review for TMLR_

### Review · Reviewer_qEFt · 2026-07-06

**Summary Of Contributions:**

This paper tackles a familiar problem in imbalanced binary tabular classification: how to generate useful synthetic minority-class samples. The authors start from a clean observation about the two dominant families of methods. Interpolation methods like SMOTE are cheap but can only fill in the convex hull of the minority points they already have, so they degrade (or simply fail) when minority data is very scarce or absent. LLM-based generators can invent samples beyond that empirical support, but they are slow because you keep calling the model. The proposed idea is to combine the two: use the LLM only to produce a small set of "seed" samples that push out the observed minority support, and then let SMOTE do the bulk of the work by interpolating inside that enlarged region. The sample-allocation accounting is laid out explicitly (Eqs. 3–6 and the worked example in Appendix A.2), which makes the procedure easy to follow.

The contribution is essentially empirical and algorithmic. The authors evaluate with two LLMs (GPT-4o-mini and Devstral-2) and SMOTE across 60 OpenML imbalanced datasets, organized around four research questions: whether the hybrid stays functional and is faster than LLM-only under few-/zero-shot conditions (Table 1); how large the runtime savings are (Figure 6, Table 7); whether the advantage grows with imbalance severity (Figure 7); and whether the hybrid generalizes more consistently from validation to test (Figure 8). They add supporting analyses on seed quality (Section 3.2), a seed-ratio sensitivity study with a Bayesian Sign Test (Section 6), and an ablation that swaps LLM seeds for SMOTE seeds (Section 7). The honest reading of the headline result is this: using the LLM only for a handful of seeds preserves whatever the LLM-only method achieves, while cutting LLM calls dramatically — and across all 60 datasets the LLM-only and hybrid methods are themselves statistically indistinguishable from plain SMOTE (Tables 5–6). So the real selling point is efficiency, plus the ability to keep working in regimes where interpolators cannot.

I think the paper has genuine merit, and I'd summarize the strengths as follows. The efficiency argument is the most convincing part, and it's convincing for a good structural reason: the LLM seeds are generated once and reused across all higher target ratios, so after seeding each additional interpolation step costs on the order of a millisecond (Table 7) against tens to hundreds of seconds for the LLM. The few-/zero-shot framing (RQ1) is well motivated — it targets a real failure mode of interpolators rather than a contrived one. The evaluation is broad for this corner of the literature (60 datasets, two LLMs, five classifiers, five target ratios), and I appreciate that the authors lean on "no free lunch" and the Bayesian Sign Test rather than cherry-picking averages; they are candid that the average differences are small and the variances large. Reproducibility is also better than typical: there's an anonymized code link, the full classifier grids (Table 4), the dataset list (Table 9), and the actual prompts (Appendix A.1).

That said, my main reservations are about the gap between what is claimed and what is shown. The abstract and contribution list promise "theoretical analyses of how and why this approach works," but I could not find any theory — Eqs. 1–6 are bookkeeping, and there are no theorems, bounds, or proofs anywhere. The performance story is also framed more favorably than the evidence supports: since all methods (SMOTE included) land within 0.6761–0.6901 mean F1 (Table 5) and the BST is dominated by "draws" (Table 6), "LLM-comparable performance" really means "no better than SMOTE either" on the ordinary regime. There's a contamination concern I'll return to below. And several of the secondary claims rest on thin evidence: RQ1 uses only four datasets with single runs (and a couple of entries that actually look like degradations or degenerate classifiers), RQ2's specific "18–31.4×" range comes from just two datasets, and the seed-quality "informativeness" claim is only really supported for SVM. Overall I see this as a useful, modest contribution whose framing needs to be brought in line with its evidence; I lean toward a major revision rather than acceptance in the current form.

**Audience:**

Yes

**Audience Explanation:**

Yes — there is a clear audience here. Researchers and practitioners working on imbalanced learning, tabular augmentation, SMOTE-style oversampling, and the newer line of work on LLMs as tabular generators would find the central finding worth knowing: you can recover most of the practical value of an LLM generator while paying for it only at the seeding step. That is a concrete, transferable design insight, and the seed-reuse runtime observation is a tidy efficiency result. The few-/zero-shot angle is also a real niche where interpolators are inapplicable and a hybrid is a sensible answer.

The contribution is incremental, and on the ordinary regime it amounts to "parity with SMOTE at much lower LLM cost than LLM-only." But TMLR explicitly accepts modest contributions that are correct, clear, reproducible, and useful, and the efficiency-at-parity result fits that description if the claims are calibrated accordingly. The scope is somewhat narrow (binary classification, two LLMs, SMOTE only, a single primary metric), but not so narrow that it falls outside TMLR's readership, and it's more than an engineering artifact because it offers a reusable recipe plus the seed-ratio analysis. The relevance would be sharper if the paper positioned itself against the broader synthetic-tabular landscape and reframed its claims around what the experiments actually establish. My "Yes" here is about audience interest and is independent of the evidence concerns above.

**Broader Impact Concerns:**

There is no Broader Impact Statement, and while I don't see serious risks, a short statement would be appropriate. Several of the 60 datasets are medical or clinical (`diabetes`, `breast-w`, `thoracic-surgery`, `primary-tumor`, `ilpd`, `sa-heart`, among others), and synthesizing minority "patient-like" records with an LLM can produce clinically implausible samples; the paper should caution against deploying such synthetic data in sensitive domains without validation. The contamination/memorization issue raised above is also partly an integrity concern, since it could lead users to overestimate how well the method generalizes. Finally, dataset licensing/consent isn't discussed; these are standard public OpenML datasets and likely low-risk, but a brief confirmation would be good. I don't identify dual-use or other unusual risks beyond these.

**Claims And Evidence:**

No

**Claims Explanation:**

I want to be clear that this "No" is about the calibration of specific claims, not about the soundness of the idea, which I think is fine.

The efficiency claim (RQ2) is the one I find genuinely well supported. Because the LLM is invoked only for seeds and those seeds are reused across target ratios, the hybrid is faster than LLM-only essentially by construction, and Table 7 / Figures 6 and 12 make the near-constant post-seeding runtime clear. My only quibble is that the precise "18.0–31.4×" figure is drawn from two datasets and should be presented as illustrative rather than as a general result.

The performance claims are where I think the paper overreaches. Table 5 puts every generation method in the 0.6761–0.6901 band, and Table 6's BST shows draw probabilities dominating most comparisons. The authors themselves say the differences aren't significant — which is the right call — but the surrounding narrative ("LLM-comparable performance," "without performance degradation") reads as a positive accuracy result when it is actually a parity result that also includes plain SMOTE. I'd ask the authors to say plainly that on standard imbalance no method dominates, and that the benefit of the hybrid is efficiency and extreme-imbalance functionality, not accuracy.

The extreme-imbalance evidence (RQ1, RQ3) is too thin to carry the weight placed on it. RQ1 is four datasets with single runs, and at least two cells cut against the "comparable" story: in zero-shot, `Birthday` reports an identical 0.4342 across the absent-baseline / LLM / hybrid columns for both LLMs (which looks like a constant-prediction classifier rather than a real result), and `Blood Transfusion` with Devstral drops from 0.5831 (LLM) to 0.4651 (hybrid). RQ3's "improvement grows with imbalance" is shown only as grouped bar heights with no confidence intervals, the authors note it isn't even monotone, and it sits uneasily next to the overall null result in Table 6.

On theory: the claim of "theoretical analyses" is simply not borne out by the manuscript, and that should be corrected regardless of anything else.

On baselines: the only competitors are SMOTE/BSM/ADASYN and the authors' own LLM pipeline. The deep generative tabular methods (CTGAN, TVAE, tabular diffusion) and the established LLM tabular generators the paper cites (GReaT, EPIC, HARMONIC) are not run, so "representational power" and "LLM-comparable" are claimed only against the authors' own LLM baseline.

On metrics and statistics: F1 is the sole primary metric, with no AUPRC, balanced accuracy, or G-mean; variance across random seeds is not reported (5-fold CV is used for configuration selection, which is a different thing); and the RQ4 variance reductions are reported without a variance-equality test. The "Achievement Rate" metric is also non-standard and conflates over- and under-fitting into one ratio.

For me to move to "Yes," I'd want: the theory claim removed and the contribution re-scoped to efficiency-at-parity plus extreme-imbalance functionality; the contamination threat addressed; RQ1 expanded to more datasets with repeated runs and reported variance (and an explanation of the odd entries); at least one additional imbalance-appropriate metric and proper significance tests for the RQ3/RQ4 claims; and either a stronger generative/established-LLM baseline or claims narrowed to exclude those comparisons.

**Requested Changes:**

**Critical for acceptance**

1. *Fix the "theoretical analysis" claim.* The abstract and contributions advertise theory that isn't in the paper (Eqs. 1–6 are notation). Either add genuine results with proofs — e.g., a precise statement about support expansion, variance, or sample complexity — or relabel this as conceptual/empirical analysis and revise the abstract, intro, and conclusion to match. Right now the stated type of contribution is inaccurate.

2. *Re-scope the performance claims to parity.* Tables 5–6 show the methods (including SMOTE) are statistically indistinguishable on the standard regime. Please state this directly and frame the contribution as efficiency plus extreme-imbalance functionality, rather than implying an accuracy gain from the LLM/hybrid machinery.

3. *Address possible benchmark contamination.* These are well-known public OpenML datasets, so GPT-4o-mini and Devstral-2 may have seen them in pretraining, which would mean the "contextual knowledge" benefit is partly memorization. I'd note that your pipeline (Appendix A.1) generates a textual report and an abstract-machine instruction sequence with jittering rather than emitting rows directly, which gives some protection against verbatim recall — but this isn't analyzed. Please add a check (e.g., results with feature-name anonymization, or on datasets unlikely to be in pretraining, or a memorization probe) and discuss the threat explicitly.

**Important but not acceptance-critical**

4. *Strengthen RQ1.* Four datasets with single runs is too little for a headline claim. Please expand the dataset count, run multiple generation seeds, report mean ± std, and explain the anomalous entries (the identical 0.4342 for `Birthday`; the 0.5831→0.4651 drop for `Blood Transfusion`-DEV).

5. *Add a competitive generative baseline, or narrow the claim.* Consider including at least one deep generative oversampler (e.g., CTGAN/TVAE) and one established LLM generator (e.g., GReaT/EPIC) under the same protocol. If that's out of scope, then please restrict the "representational power"/"LLM-comparable" language to the comparison you actually ran.

6. *Report more than F1, with proper statistics.* Add AUPRC and/or balanced accuracy or G-mean; report standard deviation over multiple seeds; use a variance-equality test for the RQ4 robustness claim and a group-level significance test (or BST) for the RQ3 trend.

7. *Fix Table 6 and specify the seed-quality protocol.* The "Win Number" rows in Table 6 are mislabeled — tie/loss rows are printed as "HYB>…" — so the win/tie/loss decomposition is currently unreadable. Separately, the uncertainty proxy in Eq. 7 doesn't say which classifier, split, or calibration is used; please specify it, and temper the informativeness conclusion to match the evidence (it's really only supported for SVM; kNN is near chance and DT is called unreliable).

**Minor / presentation**

8. *Clarify the few-/zero-shot data-modification protocol.* Give the exact per-dataset counts and the rule for moving removed minority points to validation/test, so RQ1 can be reproduced and leakage ruled out.

9. *Copy-edit.* There are recurring grammatical slips and typos (the broken opening sentence of the Introduction, "potention risk," "the results is provided," "justifies the LLM-seeds," etc.). A careful pass would help.

10. *Label Figures 1 and 2 as conceptual* so readers don't mistake the schematics for empirical results, and state the implicit assumption `r_target ≥ N_min/N_maj` behind Eqs. 3 and 5.

A few optional suggestions that aren't required: the seed-ratio sensitivity result (smallest seed ratio is both cheapest and statistically competitive) is one of the nicer findings and could be foregrounded; reporting LLM tokens/calls alongside wall-clock would make the efficiency claim more portable; and overlaying the SMOTE-seed ablation on the PCA plots (Figure 13) would visualize the support-expansion argument directly. It's also worth a sentence noting that Devstral-2 is a code model, which actually fits your code-generation pipeline even if it isn't "designed for tabular data."

---

### Review · Reviewer_bM5d · 2026-07-08

**Summary Of Contributions:**

This paper proposes a hybrid minority-class data synthesis framework that uses LLMs to generate a small number of informative seed samples and then applies interpolation to efficiently expand them, achieving LLM-comparable performance with much lower generation cost, especially under severe imbalance, few-shot, and zero-shot settings.

**Audience:**

Yes

**Audience Explanation:**

The paper proposes a simple but potentially useful hybrid direction for imbalanced tabular data synthesis: using LLMs only to generate a small number of minority-class seed samples, and then using interpolation to efficiently expand them. As the authors position the work as a foundational study of this hybrid design, it may be relevant to researchers working on tabular learning, synthetic data generation, imbalanced classification, and efficient use of LLMs.

That said, the level of interest may depend on how strongly readers value the novelty of the hybrid formulation, since the individual components—LLM-based generation and SMOTE-style interpolation—are both already known.

**Broader Impact Concerns:**

I do not see major ethical or societal risks. My main broader-impact concern is about practical positioning. The hybrid idea may already be close to engineering practice, where LLMs are used sparingly to reduce generation cost and cheaper methods are used for scaling. The paper should better clarify what is new beyond this practical intuition. In addition, the paper does not discuss recent tool-augmented or agentic LLM workflows, where LLMs can use code and statistical tools to analyze data distributions and iteratively improve synthetic samples. Such systems may become strong practical baselines for tabular data generation, and the paper should discuss how its proposed framework relates to them.

**Claims And Evidence:**

Yes

**Claims Explanation:**

The main claims are mostly supported by clear empirical evidence. The paper evaluates the hybrid LLM-seed + interpolation method on 60 imbalanced tabular datasets and shows that it achieves performance comparable to LLM-only generation while being much faster, especially in few-shot, zero-shot, and highly imbalanced settings. The runtime comparison and ablation study also support the efficiency and usefulness of LLM-generated seeds. Overall, the evidence is accurate and reasonably convincing.

**Requested Changes:**

1. The paper should provide a more detailed analysis of how the proposed method behaves across different data domains. Since the hybrid approach relies on LLM-generated seeds, its effectiveness may depend on whether the LLM has sufficient prior knowledge about the domain. For domains that are familiar to the LLM, the generated seeds may be more semantically meaningful, while for specialized or unfamiliar domains, the seeds may be less reliable or even noisy. The current paper evaluates datasets from multiple domains, but does not explicitly analyze whether domain familiarity affects seed quality or downstream performance. I suggest adding a domain-wise analysis or at least discussing this limitation more clearly.

2. Since the method itself is relatively straightforward, the paper would benefit from a broader and more up-to-date model evaluation. Without experiments on more recent LLMs, it is difficult to judge whether the reported results reflect a general property of the hybrid framework or are partly tied to the specific models used in the current experiments.

3. The theoretical analysis should be strengthened. The current paper mainly provides a formulation of the hybrid generation process, but it does not offer a rigorous explanation of why or when the hybrid method should improve imbalanced classification. A clearer theoretical discussion of the conditions under which LLM-generated seeds help, and when they may introduce noisy or out-of-distribution samples, would make the contribution more convincing.

4. The paper should discuss and compare with stronger LLM-based generation methods. The current LLM baseline appears relatively simple, relying mainly on prompting and constrained generation. However, modern LLM agents can use code and data-analysis tools to inspect feature distributions, model correlations, validate generated samples, and iteratively improve synthetic data quality. Such tool-augmented LLM methods could be much stronger baselines. Without considering them, it is hard to judge whether the hybrid method is intrinsically strong or whether the LLM-only baseline is underdeveloped.

---

### Review · Reviewer_gCLM · 2026-07-10

**Summary Of Contributions:**

The paper proposes a simple hybrid method for imbalanced tabular data: use an LLM to generate a small number of minority-class seed samples, then use interpolation to generate the rest. The main idea is to get some of the flexibility of LLM generation while avoiding the high cost of generating all synthetic samples with an LLM. The paper argues that this is especially useful when the minority class is very small or missing in training data.

**Audience:**

Yes

**Audience Explanation:**

I think this paper made some contributions to this field, and will be interested by someone who care about the efficiency of data collection on Tabular data.

**Claims And Evidence:**

Yes

**Claims Explanation:**

I think this paper made some contributions to this field, and will be interested by someone who care about the efficiency of data collection on Tabular data.

The paper has its own self-contained experiments results and analysis.

**Requested Changes:**

Strength:

A key strength of this paper is that it focuses on extreme imbalance settings, including few-shot and zero-shot cases where standard interpolation-based methods are not applicable. This makes the problem setting practically meaningful, and the results show that the proposed hybrid method can still operate in these difficult cases while achieving performance close to LLM-only generation.

Weakness:

The main motivation of the paper is efficiency, but I am not fully convinced that LLM generation cost is a serious enough bottleneck in many practical tabular settings. For small and medium datasets, generating synthetic samples with current LLMs may be affordable. Therefore, the paper should more clearly explain the benefit of the hybrid method beyond lower runtime. In particular, it is unclear whether the hybrid method improves data quality or downstream performance compared with LLM-only generation. The average F1 gains over LLM-only are small, and the paper itself reports that no method clearly dominates across all datasets. This makes the performance-efficiency trade-off unclear: the method saves LLM calls, but may also reduce the diversity and flexibility of LLM-only generation. A stronger analysis of sample quality, noise, diversity, and when interpolation helps or hurts would make the contribution more convincing.

The paper would be stronger if it framed the method not only as a cheaper alternative to LLM-only generation, but as a possible way to regularize LLM generation. However, this claim is currently under-supported. The results suggest that hybrid generation may be more stable than LLM-only generation, but the evidence is indirect and the performance gains are modest.

The paper compares against SMOTE variants and two LLM-only methods, but it may need stronger modern tabular synthesis baselines.

The proposed method seems a little bit incremental to me. The paper would be stronger if it added a more principled seed selection method, a better way to control seed quality, or a stronger theoretical explanation of when the hybrid method should help.